# A comprehensive analysis of chemical and biological pollutants (natural and anthropogenic origin) of soil and dandelion (*Taraxacum officinale*) samples

**Mieczysława Irena Boguś**[1,2,3]*, **Anna Katarzyna Wrońska**[1,2], **Agata Kaczmarek**[1,2], **Mikołaj Drozdowski**[1,2], **Zdzisław Laskowski**[2], **Anna Myczka**[2], **Aleksandra Cybulska**[2], **Marek Gołębiowski**[4], **Adrianna Chwir-Gołębiowska**[3], **Lena Siecińska**[3], **Ewelina Mokijewska**[3]

1 Museum and Institute of Zoology, Polish Academy of Sciences, Warszawa, Poland, 2 Witold Stefański Institute of Parasitology, Polish Academy of Sciences, Warszawa, Poland, 3 BIOMIBO, Warszawa, Poland, 4 Department of Environmental Analysis, Laboratory of Analysis of Natural Compounds, Faculty of Chemistry, University of Gdańsk, Gdańsk, Poland

* slawka@twarda.pan.pl

## Abstract

A range of analytical methods (GC-MS, LC-MS, voltammetry, microbiological and microscopic techniques, PCR) was used to assay a range of potential chemical and biological contaminants in soil and dandelion samples. The results provide the first comprehensive safety analysis of dandelion as a herbal product. Samples were collected from three different sites in Poland where the local population collects dandelion plants for their own consumption: Rudenka (a mountain meadow in the European Ecological Network of Natura 2000 protection area, free of agrotechnical treatments for over 30 years), Warszawa 1 (dense single-family housing with heavy traffic), and Warszawa 2 (recreation area with heavy traffic near a coal-fired heat and power plant). The assays of heavy metals and other chemical pollutants (PAHs, PCBs, dioxins, pesticides, mycotoxins) confirm that all collected soil and dandelion samples were chemically pure; however, 95 species of pathogenic bacteria were detected, including "carnivorous" *Vibrio vulnificus*, zoonotic *Pasteurella pneumotropica*, *Pasteurella canis*, *Staphylococcus pseudintermedius*, *Staphylococcus lentus* and *Francisella tularensis* as well as 14 species of pathogenic fungi and one protozoan parasite (*Giardia intestinalis*). The discovery of septicemia agents *V. vulnificus*, *Fusobacterium mortiferum* and *Rahnella aquatilis* in the soil surrounding dandelion roots and in the flowers, *G. intestinalis* in dandelion leaves and roots samples, all collected in Warsaw, is highly disturbing. This finding underlines the need for increased caution when collecting dandelion in densely populated areas with a large population of pets. Thorough washing of the harvested plants is necessary before using them for consumption, especially in the case of making salads from fresh dandelion leaves, which is becoming increasingly popular among people leading healthy and an environmentally friendly lifestyle.

**Data Availability Statement:** All relevant data are within the paper and its Supporting information files.

**Funding:** This work was supported by the Marshal's Office of the Mazowieckie Voivodeship grant RPMA.01.02.00-14-5626/16 to the Biomibo company and by the National Science Center grant 2020/39/O/NZ6/00447 to MIB. There was no additional external funding received for this study. Biomibo covered the cost of the salaries of its employees (MIB, AC-G, LS, EM), provided support in the purchase of chemicals, and made laboratory equipment available for all authors. The specific roles of the authors are articulated in the 'author contributions' section. The funders had no role in study design, data collection, analysis and interpretation, decision to publish, or preparation of the manuscript. For the purpose of Open Access, the authors haves applied a CC-BY public copyright license to any Author Accepted Manuscript (AAM) version arising from this submission.

**Competing interests:** The authors have declared that no competing interests exist.

## Introduction

Herbs have accompanied mankind since its inception. For millennia, they have been intensively used as important foodstuffs, medicines, cosmetology and beauty treatments, and have been key components of animal nutrition and breeding [1–3]. Medicinal and aromatic plants play significant roles in meeting the demands of the traditional medicine and food markets [4–6] and according to the World Health Organisation (WHO), approximately 80% of the world's population uses herbal products for therapeutic purposes [7]. The rapidly-growing interest in healthy food and the rising ecological awareness of consumers have driven a dynamic increase in sales of ecological and organic food. The healthy lifestyle trend encourages consumers to look for natural ways to support health and prevent disease. Herbal blends, herbal and functional teas, as well as herbal dietary supplements believed to strengthen immunity and help fight overweight are very popular, and the demand for herbs on the global market has been growing dynamically in recent years.

However, the safety of herbal products, both those derived from cultivated plants and from those obtained at natural sites, is coming into question. The natural environment is subject to increasing pollution as a result of the increasing combustion of fossil fuels, for heating and industrial production, and by individual human mobility and mass tourism. Such problems are exacerbated by increasing accumulation of waste in the environment, which is still insufficiently recycled.

The greatest threats of chemical contamination concern the accumulation of heavy metals, polycyclic aromatic hydrocarbons (PAHs), polychlorinated biphenyls (PCBs), and dioxins from natural and anthropogenic sources [6, 8–14]. In addition, the use of commonly-used pesticides, plant protection agents intended to control pests and to improve crop yields and quality, also poses a serious threat to the safety of picked herbs [15], as do mycotoxins, natural contaminants produced by pathogenic fungi which frequently attack growing plants and improperly-stored dried herbs [16–19].

Unfortunately, organic production, based on using only natural fertilizers while eliminating plant protection products, however, does not guarantee complete safety. Such food products can still be biologically contaminated by pathogenic microorganisms, as well as by parasitic protozoa and helminths. While studies have been performed on the microbiological safety of herbs and herbal products [20, 21], very few have examined the potential for parasitic contamination.

Most of the contamination by pathogenic microorganisms and parasites takes place from wild animals that act as intermediate or final hosts of many parasites, as well as the use of natural fertilizers (manure, slurry) and sewage sludge for cultivation [22, 23]. European Union directive 86/278/EEC [24] requires the sludge generated in the wastewater treatment process to be agriculturally reused. As part of as unique study, Zdybel and co-workers found living eggs of intestinal nematodes in 99% of samples of sewage sludge produced by municipal waste treatment plants; most samples were contaminated by the eggs of *Ascaris* spp. (95%), *Toxocara* spp. (96%) and *Trichuris* spp. (60%) [25]. The impressive survival rates of parasite eggs [26, 27] combined with their potential health hazards, such as disorders of the digestive, respiratory and nervous systems, allergies and, in the case of toxocarosis, visual disturbances leading to blindness, pose a significant threat for those consuming herbs grown on fields fertilized with sewage sludge. Parasites have also been increasingly implicated in cases of liver and pancreatic cancers and brain glioma [28, 29]. Polish environmental studies have reported the presence of *Echinococcus multilocularis* DNA in fruit, vegetable, and mushroom samples, underlining the potential risks associated with this very dangerous cosmopolitan tapeworm [30].

The aim of the current research was to comprehensively analyze a range of chemical and biological contaminants in dandelion (*Taraxacum officinale*), a cosmopolitan herbaceous perennial plant of the *Asteraceae* (Compositae) family. Dandelion is often considered a weed and grows worldwide in temperate regions with moist soils, though originally introduced from Eurasia, and can be found in parks, gardens, pastures, orchards, along roadsides, and among agricultural and horticultural crops [31, 32]. The plant has a range of uses. The Kazakh dandelion native to Kazakhstan, Kirghizia and Uzbekistan is notable for its production of high quality rubber [33, 34]. The dandelion is also an important plant in beekeeping, being a very efficient honey plant with pollens that are readily consumed by honey bees [35]. The plant also has many culinary uses: leaves are added to salads, roasted and ground roots can be used as coffee replacements, the flowers can be used to make a wine or are boiled to prepare a honey substitute. Dandelion also has a number of therapeutic properties: the roots are used to treat liver failure, dyspepsia and anorexia, while boiled leaves are used as diuretics. The anti-diabetic properties of dandelion are attributed to the bioactive components: chicoric acid, taraxasterol, chlorogenic acid, and sesquiterpene lactones [36], and the methanolic extracts of the roots have been found to have anti-proliferative effects on several human cancer cell lines, particularly against liver cancer [37, 38]. Dandelion is also believed to have a number of antibacterial and antifungal properties [1, 39].

For this medically-important plant to be used in therapeutic treatments, it must be free of chemical and biological contamination. Unfortunately there are many reports of chemical contamination in dandelion plant material and herbal teas. Previous studies have found *T. officinale* to demonstrate as much as 15-fold excess lead levels in plants growing near railway lines, accompanied by significant accumulations of PCB congeners [40, 41]. In addition, studies of individual PAH levels in several commercial herbal teas have recorded high values of naphthalene, pyrene, phenanthrene and benzo(b)fluoranthene in dandelion tea [42], and mycotoxin multicontamination by aflatoxins, ochratoxin A, zearalenone, deoxynivalenol and citrinin have been noted in the plant [43]. However, no data exists regarding potential hazardous biological contaminants such as bacteria or fungi, particularly those that are pathogenic to humans and farm animals, nor by parasitic worms and protozoa. The present study attempts to fill this gap.

## Results

### Heavy metal content in *Taraxacum officinale* and soil samples

Fig 1 shows an example voltamperogram of *Taraxacum officinale* flowers collected in Warszawa 2. Zn, Cd, Pb, Cu and Tl were determined by anodic stripping voltammetry (ASV) at hanging mercury dropping electrode (HMDE), while Co and Ni were determined by adsorptive cathodic stripping voltammetry (AdCSV) at HMDE.

Considerable differences in Cd, Co, Cu, Pb, Ni, Tl and Zn content were found between plant parts and collection sites (Table 1). The highest, and the most diverse, concentrations were found in the case of Zn (24.79–137.66 mg/kg) and Pb (14.7–80.15 mg/kg), and the lowest concentrations, with moderate differentiation, were found for Cd (0.31–4.89 mg/kg), Co (0.28–5.76 mg/kg) and Tl (0.37–23.45 mg/kg). On the other hand, the smallest variation was observed for Cu (13.56–38.14 mg/kg) and Ni (3.54–14.57 mg/kg).

The greatest amount of Cd (4.89 mg/kg) was measured in flowers collected in Warszawa 2, while the greatest amount of Co (5.76± 2.22 mg/kg) was found in soil samples from Rudenka. The highest Pb values were found in the flowers, leaves and roots of a dandelion harvested in Warszawa 2 (79.62, 77.34 and 80.15 mg/kg, respectively). Interestingly, the soil in this site contained the lowest measured Pb content (14.70 mg/kg). In each of the three sites, the highest Tl

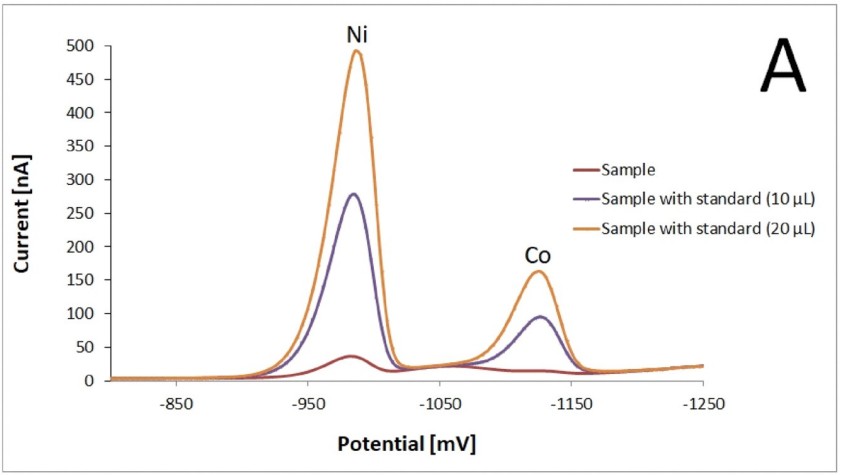

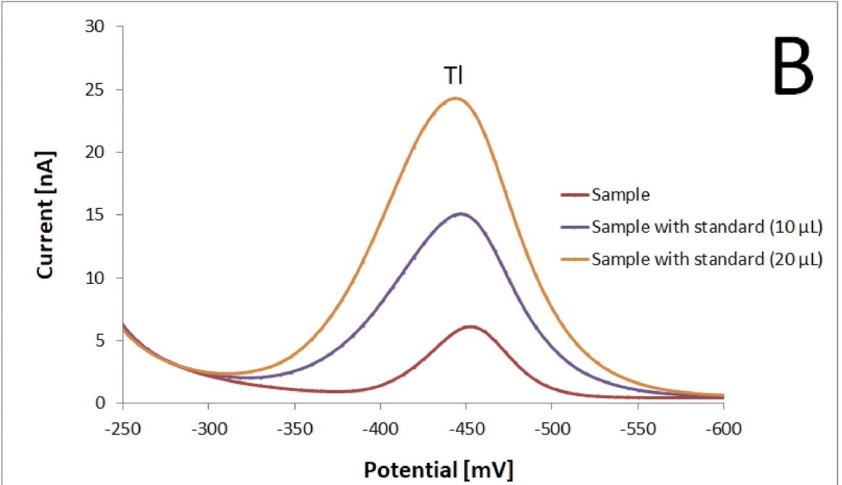

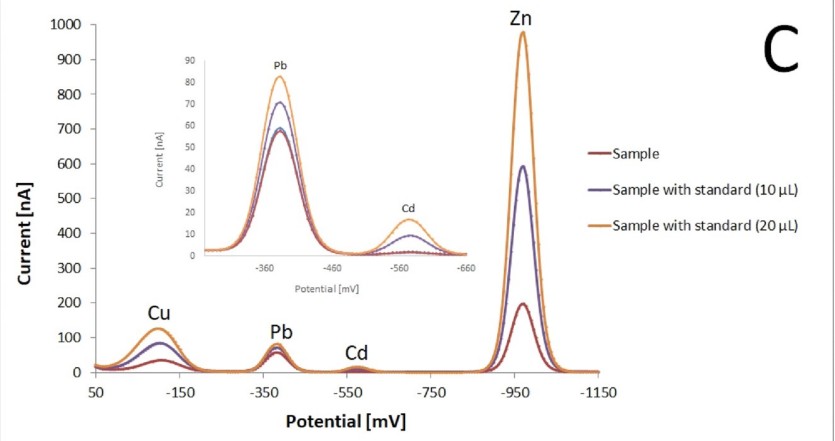

**Fig 1. Heavy metal detection in *Taraxacum officinale* flowers collected in Warszawa 2.** (A) Co and Ni determined by adsorptive cathodic stripping voltammetry (AdCSV) at hanging mercury dropping electrode (HMDE); (B) Tl determined by anodic stripping voltammetry (ASV) at hanging mercury dropping electrode (HMDE); (C) Zn, Cd, Pb and Cu determined by anodic stripping voltammetry (ASV) at hanging mercury dropping electrode (HMDE).

**Table 1. Concentration of heavy metals (mg/kg ± SD) in various parts of *Taraxacum officinale* and soil.**

| Metal | Rudenka (49.486555, 22.412092) | | | | Warszawa 1 (52.173292, 21.153900) | | | | Warszawa 2 (52.218959, 21.098110) | | | |
|---|---|---|---|---|---|---|---|---|---|---|---|---|
| | Flowers | Leaves | Roots | Soil | Flowers | Leaves | Roots | Soil | Flowers | Leaves | Roots | Soil |
| Cadmium (Cd) | 0.81±0.24 AB ab | 1.37±0.31 C ac | 1.23±0.28 E bd | 0.65±0.32 G cd | 0.47±0.13 A e | 0.66±0.19 C ef | 0.46±0.13 E f | 0.53±0.23 H | 4.89±0.14 B gh | 0.31±0.06 C gi | 0.65±0.16 E hij | 0.34±0.15 GH j |
| Cobalt (Co) | 0.28±0.02 a | 0.35±0.05 CD a | 1.02±0.28 EF a | 5.76±2.22 GH a | 0.32±0.21 e | 0.53±0.12 C e | 0.69±0.14 G e | 1.44±0.82 g | 0.24±0.07 g | 0.48±0.11 D g | 0.63±0.09 F g | 1.03±0.40 H g |
| Copper (Cu) | 23.07±3.04 AB a | 15.00±0.64 C a | 16.50±1.70 EF a | 13.56±0.21 G a | 26.97±4.12 A a | 26.23±1.34 C b | 23.79±2.24 E abc | 38.14±6.35 G abc | 29.77±4.47 B g | 19.80±1.12 C g | 23.34±2.08 F g | 16.44±0.43 G g |
| Lead (Pb) | 33.04±10.23 A a | 53.86±20.60 C ab | 43.19±6.95 E ac | 21.40±11.55 abc | 31.63±4.19 B ef | 27.34±4.21 C ef | 20.62±5.42 E e | 20.46±9.52 f | 79.62±14.29 AB g | 77.34±21.75 C h | 80.15±30.60 E i | 14.70±10.08 ghi |
| Nickel (Ni) | 5.99±0.79 A a | 3.62±0.35 C a | 7.85±1.27 E a | 13.43±4.54 GH a | 3.85±0.78 A e | 11.69±1.52 C e | 14.57±1.61 E e | 6.07±0.92 G e | 4.89±0.34 A g | 6.75±0.48 C g | 10.93±0.76 E g | 3.54±1.13 H g |
| Thallium (Tl) | 0.64±0.18 A a | 1.06±0.72 b | 0.93±0.23 E c | 12.56±0.79 G abc | 0.65±0.29 A e | 0.82±0.38 f | 0.37±0.06 EF ef | 23.45±12.47 ef | 1.14±0.35 A g | 1.00±0.39 h | 0.93±0.06 F i | 7.51±1.38 G ghi |
| Zinc (Zn) | 35.04±1.38 AB a | 29.58±3.19 C a | 24.79±1.51 E a | 37.48±2.90 G a | 37.93±3.85 A e | 66.31±5.53 E e | 54.78±5.66 E e | 76.16±7.00 G e | 39.35±2.99 B g | 60.39±8.23 C gh | 60.59±5.35 E gi | 137.66±4.58 G ghi |

N = 4–21 independent measurements. Statistically significant differences (t-test, $p < 0.05$) are marked with the same letter. Capital letters: comparisons made between samples of flowers (A, B), leaves (C, D), roots (E, F), and soil (G, H) collected at various sites (Rudenka, Warszawa 1, and Warszawa 2, respectively). Lower case letters: comparisons made between metal concentration in various parts of plants and soil surrounding their roots, Rudenka (a, b, c, d), Warszawa 1 (e, f), and Warszawa 2 (g, h, i, j). For the sake of clarity of the table, no comparisons between individual metals have been marked.

and Zn values were found in the soil samples. No regular trends were found in the measurements of Cu and Ni.

Some trends were observed regarding the concentrations of metals measured in individual plant parts and soil samples (S5 Table). No correlation of any kind was seen between the leaf and root contents. Regarding the material collected in Rudenka and Warszawa 1, positive correlations in Ni content were found between flowers and leaves ($r = 0.93$ and $r = 0.70$, respectively), while negative correlations in Co ($r = -0.89$ Rudenka), Cu ($r = -0.91$ Warszawa 1), Zn ($r = -0.84$ Warszawa 1), and Pb ($r = -0.80$ Warszawa1) content were found between flowers and roots. Zn content in the soil negatively correlated with its content in the roots ($r = -0.75$ Rudenka, $r = -0.90$ Warszawa 1), leaves ($r = -0.95$ Warszawa 2) and flowers ($r = -0.89$ Warszawa 1), while Co content in soil positively correlated with root content ($r = 0.90$ Rudenka). Cu content in soil was negatively correlated with the levels in leaves ($r = -0.86$ Rudenka, $r = -0.87$ Warszawa 2), flowers ($r = -0.70$ Rudenka, $r = -0.82$ Warszawa 1) and roots ($r = -0.90$ Warszawa 2). Pb soil content was positively correlated with leaf content ($r = 0.92$ Rudenka, $r = 0.93$ Warszawa 2). Cd soil content was negatively correlated with its content in leaves ($r = -0.95$ Warszawa 1) and flowers ($r = -0.84$ Warszawa 2); however, it positively correlated in flowers collected in Rudenka ($r = 0.70$). Similarly, the soil Co content negatively correlated with the contents in leaves ($r = -0.98$ Warszawa 2) and flowers ($r = -0.84$ Rudenka), while soil Ni content positively correlated with leaf content ($r = 0.75$ Rudenka). Tl present in soil samples was positively correlated with root ($r = 0.70$ Rudenka), leaf ($r = 0.88$ Rudenka) and flower content ($r = 0.96$ Warszawa 1), but negatively with in leaf content at Warszawa 2 ($r = -0.82$).

Interestingly, the concentration of Cd in flowers was negatively correlated with the concentration of Pb in the leaves ($r = -0.83$ Rudenka, $r = -0.72$ Warszawa 1 and $r = -0.88$ Warszawa 2, respectively). However, no other relationships in metal concentrations were found.

The biological concentration factor (BCF) values, i.e. the metal concentration in the roots vs. its concentration in soil surrounding the roots, calculated for each metal, are presented in Table 2, together with the values of two metal translocation factors (TF) demonstrating the translocation of metals from dandelion roots to leaves (TF1) and from leaves to flowers (TF2). BCF values significantly exceeding 1.0, indicating metal uptake from the soil by the roots, were found in the case of Pb (BCF range 6.78–1.45) and Cd (BCF range 4.23–1.05) while BCF values of Cu and Ni were more variable (BCF ranges 1.38–0.68 and 3.31–0.65, respectively). Low BCF values were noted for Co (0.70–0.16), Tl (0.13–0.03) and Zn (0.74–0.47), indicating a lack of

**Table 2. Cumulation of heavy metals absorbed from soil by *Taraxacum officinale* roots and translocation to leaves and flowers.**

| Metal | Rudenka(49.486555, 22.412092) | | | Warszawa 1 (52.173292, 21.153900) | | | Warszawa 2 (52.218959, 21.098110) | | |
|---|---|---|---|---|---|---|---|---|---|
| | BCF (roots/soil) | TF1 (leaves/roots) | TF2 (flowers/leaves) | BCF (roots/soil) | TF1 (leaves/roots) | TF2 (flowers/leaves) | BCF (roots/soil) | TF1 (leaves/roots) | TF2 (flowers/leaves) |
| Cadmium (Cd) | 4.23±1.39 A ab | 1.09±0.521 C a | 1.23±0.28 E b | 1.05±0.44 AB | 1.48±0.60 D c | 0.81±0.37 F c | 2.39±1.79 B f | 0.52±0.17 CD ef | 1.57±0.81 EF e |
| Cobalt (Co) | 0.16±0.04 AB a | 0.37±0.12 CD a | 0.82±0.11 E a | 0.64±0.52 A | 0.82±0.18 C c | 0.57±0.31 c | 0.70±0.23 B | 0.78±0.20 D | 0.61±0.20 E |
| Copper (Cu) | 1.25±0.07 A a | 0.88±0.06 CD a | 1.63±0.26 E a | 0.68±0.11 AB cd | 1.11±0.13 C c | 1.03±0.16 EF d | 1.38±0.11 B f | 0.87±0.11 D ef | 1.52±0.30 F e |
| Lead (Pb) | 2.95±1.69 A ab | 1.30±0.73 a | 0.88±0.35 E b | 1.45±0.81 A | 1.36±0.35 | 1.18±0.23 E | 6.78±5.15 A ef | 1.22±0.42 e | 1.11±0.25 f |
| Nickel (Ni) | 0.65±0.19 AB b | 0.47±0.03 C a | 1.65±0.09 E ab | 2.44±0.54 A c | 0.80±0.11 C c | 0.33±0.07 E c | 3.31±0.78 B e | 0.62±0.06 C e | 0.74±0.06 E e |
| Thallium (Tl) | 0.08±0.02 A a | 1.58±0.83 a | 1.15±1.45 | 0.03±0.02 A c | 2.26±0.78 C c | 0.91±0.53 c | 0.13±0.01 A ef | 1.20±0.15 C e | 1.29±0.60 f |
| Zinc (Zn) | 0.67±0.09 ab | 1.18±0.15 C a | 1.22±0.14 E b | 0.74±0.09 A c | 1.22±0.20 D c | 0.51±0.09 E c | 0.47±0.04 A e | 1.02±0.15 CD e | 0.66±0.08 E e |

BCF—Biological Concentration Factor, concentration ratio between metal absorbed from soil and accumulated in roots; TF1—Translocation Factor 1, concentration ratio between metal transferred to leaves from roots, TF2—Translocation Factor 2, concentration ratio between metal transferred to flowers from leaves. Statistically significant differences are marked with the same letter. Capital letters: comparisons made between BCFs of samples collected at various sites (A, B), TFs1 (C, D) and TFs2 (E, F), respectively. Lower case letters: comparisons made between BCF, TF1 and TF2 calculated for the samples collected in Rudenka (a, b), Warszawa 1 (c, d) and Warszawa 2 (e, f), respectively. For the sake of clarity of the table, no comparisons between individual metals have been marked.

accumulation. Pb accumulated in roots was translocated to the leaves (TF1 range 1.36–1.22), and to a lower extent, the flowers (TF2 range 1.18–0.88). Despite the lack of Tl accumulation in roots, the TF1 (2.26–1.20) and TF2 (1.29–0.91) values indicate that Tl is translocated into the leaves and flowers.

## Soil pH

The soil collected in Rudenka was slightly alkaline (pH 7.10± 0.07), while the soil collected in Warszawa 1 was neutral (pH 7.01 ± 0.04) and the soil from the site Warszawa 2 was slightly acidic (pH 6.94 ± 0.02). The statistical significance of these data is presented in Table 3.

## Lack of PAHs, PCBs, dioxins, pesticides and mycotoxin contaminants

The GC-MS analyses of the flowers, leaves, roots and soil samples taken from the three locations showed a complete absence of PCBs, dioxins and pesticides (Fig 2). Trace amounts of PAHs (below detection limit of 0.08 ng/ml) were found only in the soil sample collected in Rudenka, with their complete absence in other samples. None of the 65 tested mycotoxins (S1 Fig, S2 and S3 Tables) were detected in the tested samples.

## Microbial contaminants

Cultures of the dandelion and soil samples showed the presence of numerous aerobic and anaerobic bacteria (Table 4). The lowest number of colony forming units (CFU) was found in the case of flowers collected in Rudenka (anaerobes 14±9 CFU/mg, aerobes 21±1 CFU/mg); in contrast, anaerobic and aerobic cultures of the leaves and roots of these plants yielded 21- to 38-times higher CFUs and cultures of the soil surrounding the roots yielded 19–27 times higher numbers. No such great differences in CFU number were found at the two remaining sites, where the differences were found to range from 0.4–1.2 times (anaerobes) and 0.3–1.9 times (aerobes).

As the Biomerieux Vitek-2 system used in these studies is designed to identify clinically-significant bacteria, not all colonies obtained by plating were identified. In total, 26 species of anaerobic bacteria and 69 species of aerobic bacteria were identified in the tested samples (Table 4). Images of four homogeneous cultures of aerobic bacteria and two cultures of anaerobic bacteria, responsible for serious systemic infections and skin diseases, isolated from dandelion and soil samples, are given in Fig 3.

In all tested samples, the number of identified species of aerobic bacteria outweighed the number of anaerobic species: 10–20 species versus 2–11, respectively (Table 4). In general, samples collected in the Warszawa sites contained a more species-diverse bacterial flora than those collected in Rudenka, with the exception of aerobes in leaves.

It should be emphasized that none of the anaerobic bacterial colonies obtained from the flowers, leaves and roots of the dandelion harvested at Rudenka were clinically significant: the anaerobic *Atopobium vaginae*, *Clostridium bifermentans*, *Clostridium sporogenes* and *Fusobacterium mortiferum* found in the Rudenka soil were not detected in any parts of the plants collected there. In contrast, *Clostridium bifermentans* and *Clostridium inoculum* present in the

**Table 3. The acidity (mean pH ± SD) of the soil surrounding the *Taraxacum officinale* roots.**

| Soil acidity | Rudenka (49.486555, 22.412092) | Warszawa 1 (52.173292, 21.153900) | Warszawa 2 (52.218959, 21.098110) |
|---|---|---|---|
| pH | 7.10 ± 0.07 [A] | 7.01 ± 0.04 [B] | 6.94 ± 0.02 [AB] |

Statistically significant differences are marked with the same letter (t-test, p<0.05).

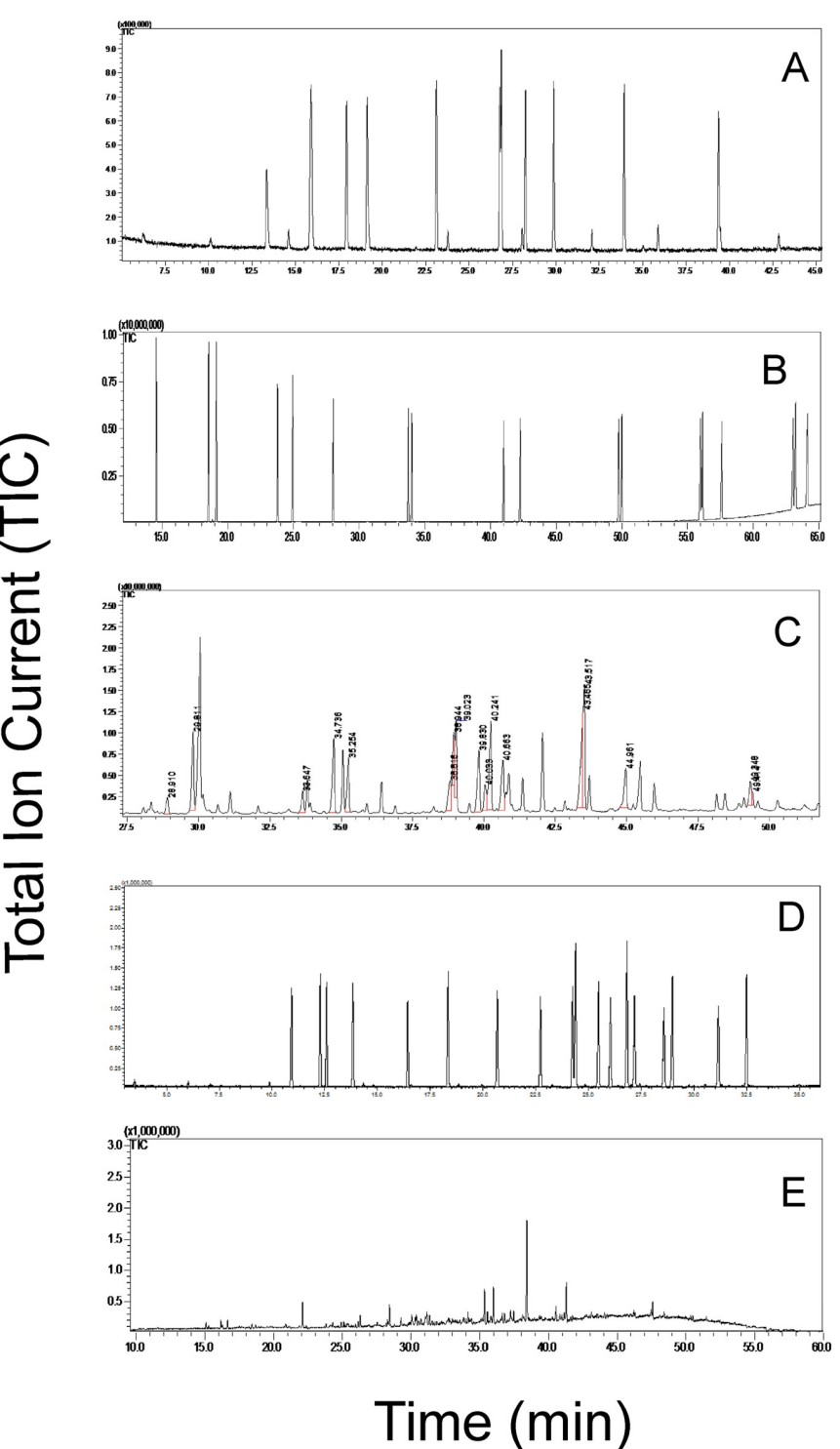

**Fig 2. The total ion current (TIC) chromatograms of: (A) PAH standards, (B) PCB standards, (C) dioxins' standards, (D) pesticide standards, (E) soil sample collected in Rudenka.** List of standards used, their retention times (RT) and limits of detection (LOD) is presented in S1 Table.

**Table 4. Anaerobic and aerobic bacteria isolated from soil and various parts of *Taraxacum officinale* plants (CFU/mg of dried sample; mean ± SD).**

| | Rudenka (49.486555, 22.412092) | Warszawa 1 (52.173292, 21.153900) | Warszawa 2 (52.218959, 21.098110) |
|---|---|---|---|
| **Flowers** | | | |
| Anaerobes: | | | |
| CFU/mg | 14 ± 9 [AB] | 390 ± 98 [A] | 385 ± 114 [B] |
| Species | Colonies not identified by Vitek-2 system | *Anaerococcus prevotii, Clostridium perfringens* and colonies not identified by Vitek-2 system | *Actinomyces odontolyticus, Anaerococcus prevotii, Clostridium* group, *Propionibacterium acnes* |
| Aerobes: | | | |
| CFU/mg | 21 ± 1 [CD] | 458 ± 154 [C] | 348 ± 139 [D] |
| Species | *Aerococcus viridans, Aeromonas salmonicida, Alloiococcus otitidis, Francisella tularensis, Kocuria kristinae, Leuconostoc mesenteroides cremoris, Methylobacterium spp, Pasteurella pneumotropica, Pediococcus pentosaceus, Photobacterium damselae, Sphingomonas paucimobilis, Staphylococcus lentus, Staphylococcus saprophyticus, Staphylococcus vitulinus,* | *Acinetobacter lwoffii, Aeromonas salmonicida, Agrobacterium tumefaciens, Alloiococcus otitidis, Bacillus cereus, Bacillus megaterium, Cronobacter sakazakii, Enterococcus casseliflavus, Helcococcus kunzii, Leuconostoc mesenteroides cremoris, Lysinibacillus fusiformis, Pantoca sp., Pasteurella testudinis, Pseudomonas luteola, Sphingomonas paucimobilis, Staphylococcus hominis, Staphylococcus vitulinus, Staphylococcus warneri* | *Acinetobacter lwoffii, Alicyclobacillus acidoterrestris, Bacillus cereus, Francisella tularensis, Pseudomonas luteola, Rahnella aquatilis, Sphingomonas paucimobilis, Staphylococcus aureus, Staphylococcus lentus, Staphylococcus xylosus* |
| **Leaves** | | | |
| Anaerobes: | | | |
| CFU/mg | 299 ± 185 | 162 ± 76 [E] | 385 ± 114 [E] |
| Species | Colonies not identified by Vitek-2 system | *Atopobium vaginae, Clostridium bifermentans, Fusobacterium varium, Veillonella spp* | Colonies not identified by Vitek-2 system |
| Aerobes: | | | |
| CFU/mg | 396 ± 107 [F] | 173 ± 25 [FG] | 348 ± 139 [G] |
| Species | *Aerococcus viridans, Alloiococus otitidis, Bacillus cereus, Bacillus pumilus, Gemella sanguinis, Kocuria rhizophila, Leuconostoc mesenteroides dextranicum, Pantoea spp., Pasteurella testudinis, Staphylococcus vitulinus, Sphingomonas paucimobilis, Staphylococcus lugdunensis, Staphylococcus pseudintermedius, Staphylococcus sciuri* | *Acinetobacter lwoffii, Aeromonas salmonicida, Bacillus cereus, Gemella morbillorum, Methylobacterium spp., Pantoea spp., Pseudomonas oryzihabitans, Sphingomonas paucimobilis, Staphylococcus lugdunensis, Staphylococcus pseudintermedius* | *Aerococcus viridans, Alloicoccus otitidis, Bacillus cereus, Bacillus megaterium, Kocuria rosea, Pantoea spp., Sphingomonas paucimobilis, Staphylococcus aureus, Staphylococcus lentus, Staphylococcus vitulinus* |
| **Roots** | | | |
| Anaerobes: | | | |
| CFU/mg | 535 ± 124 [JK] | 273 ± 65 [HJKL] | 481 ± 80 [IL] |
| Species | Colonies not identified by Vitek-2 system | *Atopobium vaginae, Clostridium baratii, Clostridium bifermentans, Clostridium glycolicum, Clostridium innocuum, Clostridium paraputrificum, Clostridium perfringens, Clostridium sordelli* | *Atopobium vaginae, Clostridium baratii, Clostridium bifermentans, Clostridium paraputrificum, Clostridium perfringens, Clostridium tertium, Eggerthia catenaformis, Fusobacterium mortiferum, Fusobacterium necrophorum, Fusobacterium nucleatum, Parabacteroides distasonis* |
| Aerobes: | | | |
| CFU/mg | 467 ± 55 [MN] | 146 ± 20 [HM] | 152 ± 12 [IN] |
| Species | *Alloiococcus otitidis, Bacillus cereus, Bacillus megaterium, Brevibacillus choshinensi, Francisella tularensis, Leuconostoc mesenteroides cremori, Ochrobactrum anthropi, Sphingomonas paucimobilis, Staphylococcus pseudintermedius, Virgibacillus pantothenticus* | *Acinetobacter baumannii, Acinetobacter haemolyticus, Aeromonas sorbia, Alloiococcus otitidis, Bacillus cereus, Enterococcus columbae, Escherichia coli, Francisella tularensis, Gardnella vaginalis, Leuconostoc mesenteroides cremoris, Methylobacterium spp., Neisseria cinerea, Oligella ureolutica, Pantonea spp, Pasteurella canis, Pseudomonas luteola, Sphingomonas paucimobilis, Staphylococcus aureus, Staphylococcus paucimobilis, Staphylococcus pseudintermedius* | *Aeromonas salmonicida, Aeromonas sobria, Alloiococcus otitidis, Bacillus cereus, Enterococcus columbae, Escherichia coli, Francisella tularensis, Leuconostoc mesenteroides cremoris, Moraxella grup, Oligella ureolytica, Pasteurella testudinis, Pseudomonas oryzihabitan, Sphingomonas paucimobilis, Staphylococcus aureus, Staphylococcus lugdenensis, Staphylococcus pseudintermedius* |

*(Continued)*

**Table 4.** (Continued)

| | Rudenka (49.486555, 22.412092) | Warszawa 1 (52.173292, 21.153900) | Warszawa 2 (52.218959, 21.098110) |
|---|---|---|---|
| **Soil** | | | |
| Anaerobes: | | | |
| CFU/mg | 349 ± 115 [O] | 489 ± 75 [R] | 268 ± 39 [PR] |
| Species | *Atopobium vaginae, Clostridium bifermentas, Clostridium sporogenes, Fusobacterium mortiferum* and colonies not identified by Vitek-2 system | *Actinomyces odontolyticus, Bifidobacterium spp, Campylobacter ureolyticus, Clostridium bifermentas, Clostridium difficile, Clostridium innocuum, Clostridium subterminale, Fusobacterium mortiferum, Fusobacterium necrophorum, Fusobacterium varium, Propionibacterium acnes, Terrisporobacter glycolicus* | *Atopobium vaginae, Clostridium perfringens* and colonies not identified by Vitek-2 system |
| Aerobes: | | | |
| CFU/mg | 563 ± 120 [O] | 382 ± 140 [S] | 677 ± 113 [PS] |
| Species | *Aeromonas salmonicida, Alloiococcus otitidis, Bacillus cereus, Bacillus farraginis, Bacillus megaterium, Escherichia coli, Gemella haemolysans, Leclercia adecarboxylata, Methylobacterium spp, Pasteurella pneumotropica, Sphingomonas paucimobilis, Staphylococcus lugdunensis, Staphylococcus pseudintermedius, Staphylococcus vitulinus, Streptococcus anginosus* | *Acinetobacter lwoffii, Aeromonas veronii, Bacillus cereus, Brevundimonas diminuta, Cedecea lapagei, Chromobacterium violaceum, Enterococcus durans, Pantoea spp., Pseudomonas luteola, Pseudomonas oryzihabitans, Sphingomonas paucimobilis, Sphingobacterium spiritivorum, Staphylococcus lugdunensis, Staphylococcus pseudintermedius, Vibrio vulnificus* | *Acinetobacter haemolyticus, Alloiococcus otitidis, Bacillus megaterium, Haemophilus parainfluenzae, Lactococcus garvieae, Pasteurella testudinis, Photobacterium damselae, Pseudomonas luteola, Sphingomonas paucimobilis, Staphylococcus aureus, Staphylococcus lugdunensis* |

CFU—colony forming units. Statistically significant differences are marked with the same letters (t-test, $p < 0.05$). Comparisons made between samples of flowers (A-D), leaves (E-G), roots (H-N), soil (O-S) collected at various sites (Rudenka, Warszawa 1, Warszawa 2).

Warszawa 1 soil was also found in roots and leaves of dandelion collected there, and *Atopobium vaginae* and *Clostridium perfringens* present in the soil from Warszawa 2 were also present in the root and leaf samples of the dandelion harvested at this site.

The most common pathogenic aerobes were *Sphingomonas paucimobilis*, a rare infectious agent (detected in all samples), *Bacillus cereus*, producing toxins (absent only in Rudenka flowers and Warszawa 2 soil) and *Alloicoccus otitis* involved in middle ear infections (absent in Warszawa 2 flowers and Warszawa 1 leaves and soil) (Table 4). The presence of *Escherichia coli*, commonly found in the gut of humans and warm-blooded animals, was confirmed only in the Rudenka soil and Warszawa 1 roots, while *Staphylococcus aureus*, a major human pathogen, was detected in all Warszawa 2 samples and Warszawa 1 roots. Of the zoonotic pathogens, *Pasteurella pneumotropica* was found in Rudenka flowers and soil, *Pasteurella canis* only in Warszawa 1 roots, *Staphylococcus pseudintermedius* in all Rudenka and Warszawa samples except flowers and *Staphylococcus lentus* in Rudenka flowers and Warszawa 2 flowers and leaves. *Francisella tularensis*, present in all root samples, as well as in Rudenka and Warszawa 2 flowers, was absent in all soil and leaf samples.

The septicemia agents *Vibrio vulnificus* and *Fusobacterium mortiferum* were detected in Warszawa 1 soil, while *Rahnella aquatilis* in Warszawa 2 flowers. Pneumonia-inducing *Clostridium baratii* was found in Warszawa 2 roots and gangrene-inducing *Clostridium perfrigens* in Warszawa 1 roots. In addition, *Haemophilus parainfluenzae*, known to cause a variety of severe infections, was found in Warszawa 2 soil, while *Staphylococcus lugdunensis* an infective agent of various diseases, was detected in leaves and soil collected in the Rudenka and Warszawa 1 sites.

# Aerobes:

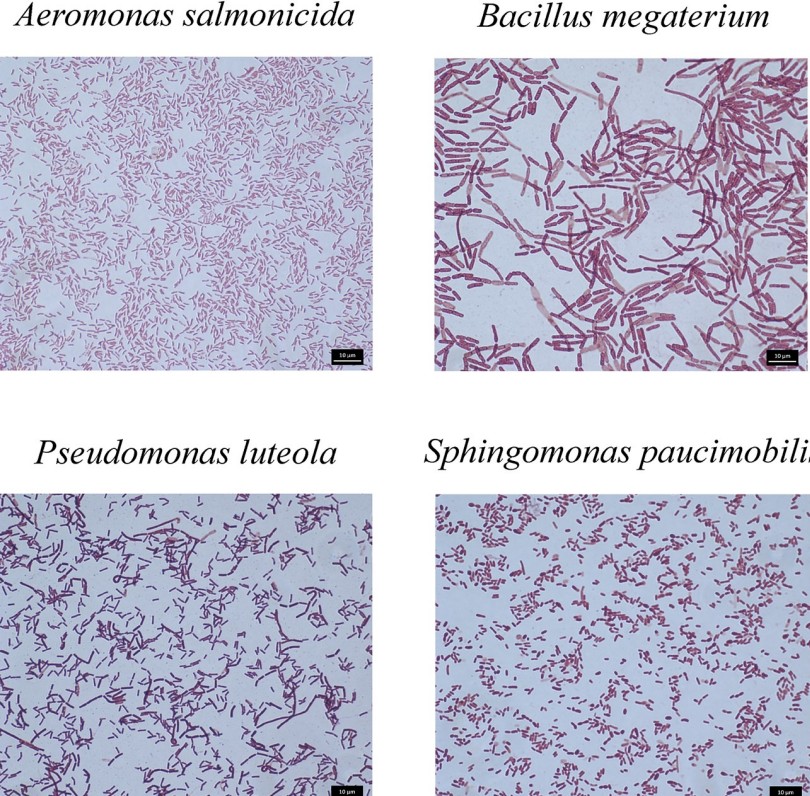

*Aeromonas salmonicida*          *Bacillus megaterium*

*Pseudomonas luteola*          *Sphingomonas paucimobilis*

# Anaerobes:

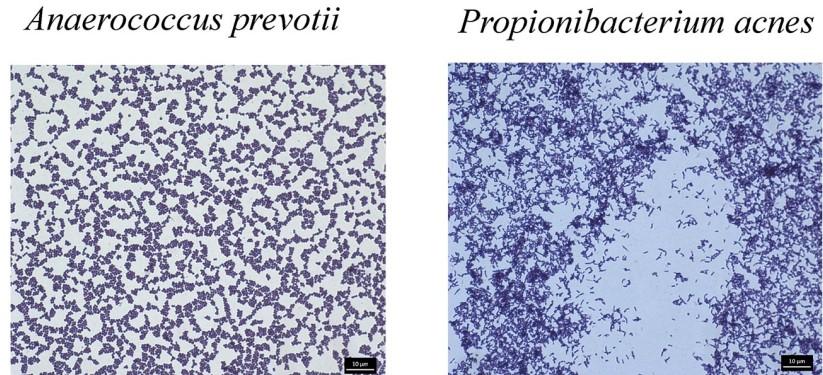

*Anaerococcus prevotii*          *Propionibacterium acnes*

**Fig 3. Examples of pathogenic bacteria detected in *Taraxacum officinale* and soil samples.** Scale bars: 10 μm.

Table 5 shows the result of mycological analysis of collected samples. The highest number of CFU was found in Warszawa 2 soil (9838±1062 CFU/mg) and the lowest in Warszawa 2 flowers (212±109 CFU/mg); however, the total number of detected fungal species, approximately 14, was significantly smaller than that of the bacteria. The macroscopic analysis of

**Table 5. Fungi isolated from soil and various parts of *Taraxacum officinale* plants.**

| | Rudenka (49.486555, 22.412092) | Warszawa 1 (52.173292, 21.153900) | Warszawa 2 (52.218959, 21.098110) |
|---|---|---|---|
| **Flowers:** | | | |
| CFU/mg | 740 ± 294 | 558 ± 270 [D] | 212 ± 109 [G] |
| Identified species | *Aspergillus niger, Penicillium sp.* | *Alternaria alternata, Aspergillus fumigatus, Aspergillus niger, Cladosporium sp., Penicillium sp.* | *Alternaria alternata, Aspergillus niger, Penicillium sp.* |
| **Leaves:** | | | |
| CFU/mg | 1573 ± 1492 | 1045 ± 648 [E] | 281 ± 137 [H] |
| Identified species | *Alternaria alternata, Aspergillus fumigatus, Fusarium sp., Penicillium sp.* | *Alternaria alternata, Aspergillus fumigatus, Aspergillus sp., Cladosporium sp, Mucor sp., Penicillium sp., Rhizopus sp.* | *Alternaria alternata, Asperillus sp., Mucor sp., Penicillium sp.* |
| **Roots:** | | | |
| CFU/mg | 1692 ± 1687 | 1476 ± 1564 [F] | 287 ± 87 [I] |
| Identified species | *Alternaria alternata, Fusarium sp., Paecilomyces sp., Penicillium sp., Trichoderma viride* | *Alternaria alternata, Aspergillus terreus, Chaetomium sp., Cladosporium sp., Fusarium sp., Mucor sp., Penicillium sp.* | *Aspergillus niger, Penicillium sp., Rhizomucor sp.* |
| **Soil:** | | | |
| CFU/mg | 459 ± 226 [AB] | 3872 ± 698 [ACDEF] | 9838 ± 1062 [BCGHI] |
| Identified species | *Penicillium sp., Trichoderma viride* | *Mucor sp., Penicillium sp.* | *Mucor sp.* |

CFU (colony forming units) are presented as mean CFU/mg of dried sample ± SD. Statistically significant differences are marked with the same letters (t-test, p range from p = 0.0494 to p<0.0001).

fungal colonies and microscopic images of fungal structures only allowed precise species identification in five cases (Fig 4). *Aspergillus niger* was identified in all flower samples and in Warszawa 2 roots, *Aspergillus fumigatus* in Warszawa 1 flowers and leaves as well as Rudenka leaves, while *Aspergillus terreus* was only recorded in Warszawa 1 roots. *Alternaria alternata* was found in all leaf samples, both Warszawa flowers, Rudenka and Warszawa 1 roots, while *Trichoderma viridae* was found only in Rudenka roots and soil. For the fungi belonging to genera *Penicillium*, *Mucor*, *Rhizomucor*, *Fusarium*, *Rhizopus*, *Cladosporium* and *Chaetomium* it was not possible to determine their species. *Penicillium spp.* was present in all flower, leaf and root samples, as well as in Rudenka and Warszawa 1 soil. *Mucor spp.* was found in Warszawa 1 leaves, roots and soil, as well as in Warszawa 2 leaves and soil. *Fusarium spp.* was detected in Rudenka leaves and roots, as well as in Warszawa 1 roots, and *Rhizopus spp.* in Warszawa 1 leaves. *Paecilomyces spp.* was found in Rudenka roots. *Rhizomucor spp.* was found Warszawa 2 roots, *Chaetomium spp.* in Warszawa 1 roots, and *Cladosporium spp.* in all dandelion parts in Warszawa 1 (Table 4).

## Parasite contamination

All the collected samples were tested for the presence of parasite eggs and DNA. Neither helminth eggs nor DNA (tapeworms, nematodes, and flukes) were found in any of them (Fig 5A and 5B). In the Warszawa 2 site, the presence of the protozoan parasite *Giardia intestinalis* DNA was confirmed in dried leaves and roots (Fig 5C). The 489 bp partial sequence of triosephosphate isomerase (tpi) gene obtained from dried dandelion leaves (GenBank accession number OM964640.1), was 99.8% similar to seven *Giardia intestinalis* (assemblage C) sequences obtained from dogs in Slovakia, Brazil, Japan and Poland (GenBank accession numbers: LC437516.1, KT728510.1, KT728509.1, KT728507.1, MZ160223.1, EU781006.1, OM964641.1).

**Fig 4. Fungi detected in *Taraxacum officinale* and soil samples.** (A) macroscopic view of fungal colony, (B) microscopic image of the fungus. Scale bars: 20 μm.

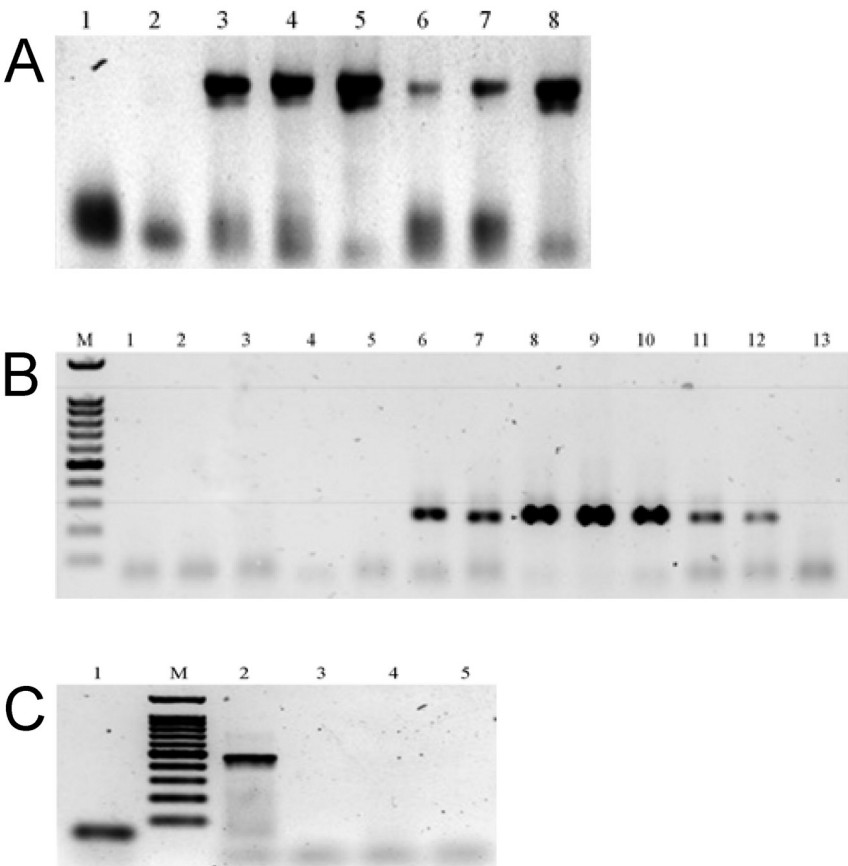

**Fig 5. Detection of parasites' DNA in *Taraxacum officinale* samples.** (**A**) Helminth Test, 1—H₂O, 2—dried dandelion leaves (Rudenka), 3—*Oxyuridae* sp. (eggs), 4—*Echinococcus* sp. (adult), 5—*Taenia* sp. (eggs), 6—*Trichuris* sp. (eggs), 7—*Fasciola hepatica* (adult), 8—*Toxocara cati* (adult); (**B**) Helminth Test, M—Marker DNA, 1- dried dandelion leaves (Warszawa 1), 2—dried dandelion leaves (Warszawa 2), 3—dried dandelion flowers (Warszawa 2), 6—*Echinococcus* sp. (adult), 7—*Toxocara cati* (adult), 8—*Dipylidium caninum* (adult), 9—*Fasciola hepatica* (adult), 10—*Taenia taeniaeformis* (eggs), 11—*Toxocara canis* (eggs), 12—Oxyuridae sp. (eggs), 13—H₂O; (**C**) Giardia Test, 1—H₂O, M—Marker DNA, 2—dried dandelion leaves (Warszawa 2), 3—dried dandelion flowers (Warszawa 2), 4—dried dandelion leaves (Rudenka), 5—dried dandelion flowers (Rudenka). DNA marker—100 bp DNA Ladder (Novazym).

## Discussion

In the present study, a comprehensive examination was made of a wide range of possible contaminants of dandelion collected from three different sites. One was a mountain meadow in a small village, forming part of the Natura 2000 European Ecological Network, close to the Polish-Ukrainian border. For over 30 years, this meadow has not been subjected to agrotechnical treatments, apart from mowing once a year, and has not been used for grazing livestock; it is commonly visited by wild animals such as deer, roe deer, hares and foxes, with some visits by bears and wolves. The other two sampling sites for testing were located in Warsaw, the Poland's capital city. Sampling site Warszawa 1 was situated in an area with dense single-family housing, without any industry but with heavy traffic, while Warszawa 2 was in an extensive recreation area on the Vistula River bank, also with heavy traffic and located near to the coal-fired heat and power plant providing heat for 25% of the facilities in Warsaw. At all of these sites, the dandelion leaves are collected by the local population for salads, flowers to prepare honey substitutes and roots for medication, and for feeding small pets kept by children, such as rabbits and guinea pigs.

All the collected plant and soil samples were tested for heavy metal content using the anodic stripping voltammetry (ASV) method at hanging mercury dropping electrode (HMDE); this method allows a simultaneous evaluation of several metals including Pb, Cu, Cd Zn and Tl, even at very low levels of parts-per-million (ppm) or even parts-per-billion (ppb). In addition to being simple and selective, this approach is also much less expensive than spectroscopic techniques [44, 45]. ASV is a two-step measurement. In the first step, called *amalgamation*, the metal ions present in the test solution are deposited on the surface of the mercury electrode, while in the second step, all the deposited ions are anodically stripped. ASV probes the species really existing in solutions which is a great advantage over atomic spectroscopy, in which the species present in solution are transferred to a plasma where they are present as atoms [45]. Ni and Co were also assayed using adsorptive cathodic stripping voltammetry (AdCSV) at hanging mercury dropping electrode (HMDE). The AdCSV technique is based upon the adsorptive accumulation of the metal ion complex with a suitable ligand at the electrode. In both methods, the working electrode was a mercury electrode; it is easy to prepare, and offers sensitive and reproducible results, with fast and simple electrode kinetics [44, 46]. Presented in this work voltammetry measurements of Zn, Pb, Cd, Ni and Cu in *T. officinale* were similar to those obtained by other authors using spectroscopic methods [47–51]. However, samples demonstrated a much higher Cu content than that reported by Habh and co-workers [52].

According to Minister of Agriculture and Rural Development regulations, the concentrations of heavy metals polluting the light agricultural soils in organic farming in Poland must not exceed 50 mg/kg (Pb), 0.75 mg/kg (Cd), 30 mg/kg (Ni), 100 mg/kg (Zn) and 30 mg/kg (Cu) [53]. Data in Table 1 clearly show that the Pb, Cd and Ni concentrations were significantly below these limits in all soil samples; however, the Cu content in the Warszawa 1 soil sample (38.14 mg/kg) slightly exceeded the limit for light soils but was below the limit for medium heavy and heavy soils (50 and 75 mg/kg, respectively) [53]. Similarly the Zn content exceeded the levels allowed for the light soil in Warszawa 2 (137.66 mg/kg), but was below the limit in the other soil samples. The maximum permitted Zn content is 200 mg/kg in medium heavy soils and 300 mg/kg in heavy soils [53]. As the soil is medium heavy at both Warszawa sites and heavy in Rudenka, our findings confirm the purity of all soil sample collection sites particularly compared with soil and plants sampled in the industrial region of Silesia in Poland [48, 54].

The presence of slightly elevated Zn levels in soil and dandelion samples collected in Warsaw may reflect the impact of road traffic and power plant emissions [49, 55, 56]. Literature data indicate strong Zn absorption by plant roots and high mobility inside plant [57]; however, our data only confirm these observations to a certain extent, and only in terms of Zn translocation from roots to leaves (TF1). The BCF accumulation index was in each case significantly below the 1.0 value. In contrast, high BCF values were observed for Pb and Cd, 6.78 and 4.23 respectively; these indicate intensive uptake of these metals from the soil by dandelion roots. The absorbed metals were translocated to both leaves and flowers.

However, the process of Tl absorption from soil is unclear. Its very low BCF coefficients suggest no absorption, while TF1 and TF2 values exceeding 1.0 suggest active Tl transfer from the roots to the leaves and from the leaves to the flowers. Further research is needed to clarify this issue as the accumulation of the metals in plants and their distribution inside plant tissues might be affected by numerous environmental and physiological factors.

No chemical contaminants (PAHs, PCB, dioxins, pesticides) were found in any samples of dandelion flowers, leaves or roots collected at the three different sites, suggesting the dried plants are safe for consumption in this respect. It is not surprising that unidentifiable traces of PAHs were detected in the soil from Rudenka since this is located in the oil-bearing flysch Carpathians, with a geological structure typical of young fold mountains, in which tectonic

engagement results in good hydrocarbon traps [58, 59]. The Rudenka site is about 20 km from shafts and oil and gas deposits that are currently being exploited.

Unfortunately, the chemical purity of the samples was not accompanied by equal biological purity: a total of 95 pathogenic bacterial species (26 anaerobes and 69 aerobes) were identified. It should be noted that while no anaerobic pathogenic bacteria were found in the plant samples collected in Rudenka, they were present in the plants collected in Warsaw. The *Propionibacterium acnes* (present name *Cutibacterium acnes*) anaerobic bacteria, believed to play a significant role in the pathogenesis of acne vulgaris and prosthetic joint infections [60, 61], were identified only in Warszawa 1 soil and Warszawa 2 flowers, i.e. in places with high population density. In contrast, anaerobic *Atopobium vaginae*, an important component of the complex abnormal vaginal flora in bacterial vaginosis [62, 63], was present in samples from all tested places. In our samples, the most common aerobic bacterial species were *Sphingomonas paucimobilis*, a rare infectious agent which may cause both nosocomial and community-acquired infections, albeit rarely [64, 65], *Bacillus cereus*, producing the heat-stable toxin cereulide known to cause food-borne illness [66, 67], and *Alloiococcus otitidis*, a common bacterium of the human ear; however, while it has been found in a high percentage of middle ear effusions in children, it is still not clear whether this bacterium really is a pathogen [68].

Most of the tested samples demonstrated *Staphylococcus pseudintermedius*, a canine opportunistic pathogen, one of the important human pathogens of zoonotic origin known to cause wound and skin infections [69, 70] and *Francisella tularensis* a highly infectious coccobacillus transmitted to humans by a number of different routes including ingestion of contaminated food [71, 72]. In addition, they were found to harbor *Staphylococcus lentus*, an animal pathogen isolated from rodents, chickens, mammals, farm soil and water, known to cause endocarditis septic shock, urinary tract infection, wound infections, endophthalmitis, and pelvic inflammatory disease in humans [73]. In contrast, *Pasteurella pneumotropica* responsible for rodent pasteurellosis known as a causative agent in infected humans of endocarditis, pneumonia, meningitis, osteomyelitis, arthritis, septicemia and epidural abscess [74, 75], was detected only in Rudenka flowers and soil, i.e. in an area with a large number of voles. Similarly, *Pasteurella canis*, known as a part of the normal flora of healthy domestic animals but a major pathogen of infections caused by dog bites [76, 77], was found only in dandelion roots collected at the Warszawa 1 site, where there is a large population of dogs and cats. Most disturbingly, the "carnivorous" *Vibrio vulnificus*, causing wound infections and septicaemia [78, 79], *Fusobacterium mortiferum* and *Rahnella aquatilis* causing bacteremia [80, 81] were also identified in the Warsaw samples.

It is difficult to account for the presence of *Pasteurella testudinis*, which is a pathogen of desert turtles [82, 83], in the samples at all three sites, i.e. in the mountains and in Warsaw, because no turtles are present in these sites. A similar problem applies to *Aeromonas salmonicida* a fish pathogen causing a disease known as furunculosis [84, 85], as there are no breeding ponds in the vicinity of these sites; however, one report concerning isolation of this bacterium from human blood is available [86].

Herbal plants have been used in traditional medicine for centuries due to their ability to synthesize numerous antibacterial and antifungal compounds, which probably evolved as defenses against infection and/or predation [87]. Dandelion has long been known to have antimicrobial potential, and it has had a place in traditional medicine worldwide for centuries, being prescribed for malaria, tuberculosis, cough, and various bacterial infections [88, 89].

Research on the antimicrobial properties of dandelion has gained particular relevance in the face of antibiotic resistance and various extracts from this plant have been tested on bacterial and fungal strains affecting humans, animals and plants, both to confirm its traditional usage and expand its known uses [1]. The list of pathogens tested so far includes only five

bacterial species found in our samples: *Bacillus cereus*, *Bacillus pumilus*, *Escherichia coli*, *Chromobacterium violaceum* and *Propionibacterium acnes*. The test results were clearly positive only in the case of the two latter species: the extracts from *T. officinale* were active against these pathogens [90–95].

The tests carried out so far on pathogenic fungi only clearly indicate that the dandelion extracts inhibit *Alternaria alternata* [96, 97], while the results concerning *Aspergillus niger* and *Aspergillus fumigatus* were inconclusive [1, 94, 98–101]. The sensitivity of the other fungal species present in our samples to dandelion extracts remains unknown.

Assuming that no trace of mycelia development was observed in the samples, it is possible that fungal spore germination was halted, perhaps by the antifungal substances present in dandelion plants. This inhibition of spore germination and mycelium growth might explain our failure to find any of the 65 mycotoxins tested, despite the presence of significant amounts of spores of mycotoxin-producing fungi in the samples. It is known that *Aspergillus niger* produces fumonisin B2 and ochratoxin A [102–104], *Aspergillus fumigatus* produces fumagillin and gliotoxin [105–107], *Aspergillus terreus* produces territrem A and B [108], while *Alternaria alternata* produces alternariol, alternariol-9-methyl ether and tenuazonic acid [109]. It has also been known that trichothecene mycotoxins produced by *Stachybotrys chartarum*, the etiologic agent of stachybotryotoxicosis, can accumulate to very high levels in the conidia [110, 111]; however, it remains unclear whether the spores of the fungi present in our samples contained mycotoxins, and if so, whether they were present in detectable amounts. Further research is needed to address this problem.

Although the presence of protozoan cysts, helminth eggs and invasive stages of intestinal parasites on fruits and vegetables and in the soil is mainly associated with tropical countries [112–115], their presence has also been recorded in Poland [30, 116]. Disease-causing intestinal parasites are traditionally detected by microscopic examination of stool samples which is still considered the "gold standard" for the diagnosis; however, owing to their greater sensitivity and accuracy, molecular techniques such as conventional PCR, RT-PCR, nested PCR, LAMP and other modern technologies are being increasingly introduced into diagnostics with some success [117–122].

Research presented here was performed using multiplex PCR tests. The analysis itself employed self-made primers based on partial 18S rRNA species-specific sequences deposited in GenBank; these allow the detection of DNA from tapeworms, nematodes, flukes and *Giardia* species reported in Poland. The 18S rRNA sequences are widely used for primer development as they are highly polymorphic and conserved, allowing species-level separation, and have proven effective for establishing phylogenetic relatedness among species [119]. In the material, only the DNA of *Giardia intestinalis* was found in samples of dandelion leaves and roots taken from the Warszawa 2 stand; this area is often visited by walkers with dogs.

Traditionally, when thinking on herbal plant contamination, everyone is more focused on chemical than biological contaminations, however the progressive deindustrialization of Europe has significantly reduced the threats from the accumulation of heavy metals, polycyclic aromatic hydrocarbons, polychlorinated biphenyls, and dioxins [122]. Increasing acreages of organic farming also have reduced the risks from the use of pesticides [123]. This is confirmed by this research, which showed a very high chemical purity of the collected plant samples and the soil directly surrounding the roots of dandelion. On the other hand, the risks associated with biological contamination of herbs, especially in densely populated territories and with a large population of domestic animals, are frequently underestimated. The discovery of dangerous bacteria *Vibrio vulnificus*, *Fusobacterium mortiferum* and *Rahnella aquatilis* as well as the protozoan parasite *Giardia intestinalis* in tested samples is highly disturbing. These hazardous pathogens have been detected in the recreational area and in the vicinity of single-family

housing estates in Warsaw, the capital of Poland, a metropolis with a population of nearly two million.

## Conclusions

A broad analysis of chemical and biological contaminations was performed on soil and dandelion samples using a wide range of analytical methods (GC-MS, LC-MS, voltammetry, microbiological and microscopic techniques, PCR). The findings provide the first comprehensive safety analysis of a herbal product. The presented analysis of the heavy metal content and the presence of other chemical pollutants (PAHs, PCBs, dioxins, pesticides, mycotoxins) indicate that all collected soil and dandelion samples were of high chemical purity. In turn, biological analyses showed the presence of 95 species of pathogenic bacteria, 14 species of pathogenic fungi and one protozoan parasite (*Giardia intestinalis*); this underlines the need for increased caution when collecting dandelion in densely populated areas with a large population of pets. Thorough washing of the harvested plants is necessary before using them for consumption, especially in the case of making salads from fresh dandelion leaves, which is becoming increasingly popular among people leading healthy and an environmentally friendly lifestyle. The use of heat treatment when preparing traditional infusions and flower preserves should significantly reduce any risk of infection.

## Materials and methods

### Plant and soil sample collection

Herbs were collected from two locations in Warsaw: Warszawa 1 (52.173292, 21.153900) situated in area of heavy traffic without any industry, Warszawa 2 (52.218959, 21.098110) in a recreation area with traffic near to the coal-fired Siekierki heat and power plant with a thermal power of 2068 MW and an electric power of 620 MW. A third set of samples was also taken from Rudenka (49.486555, 22.412092), part of the Natura 2000 European Ecological Network, located in a sparsely-populated region 30 km from the Polish-Ukrainian border. The sample site itself is a meadow which has lain fallow for over 30 years, during which time, it has not been subjected to any agrotechnical treatments nor used for breeding livestock. No permits were required for the collection of samples because the two collection sites in Warsaw (Warszawa 1 and Warszawa 2) are open to everyone without any restrictions and the third collection site in Rudenka is located on the private property of the corresponding author of this work (MIB). Whole *T. officinale* plants were harvested in May 2018 just after flowering began. The samples were divided into parts (flowers, leaves, roots) and air dried in a cool, shaded and airy room. The soil surrounding the dandelion roots was also collected and air dried.

### Preparation of soil samples for pH determination

The soil samples were shredded in a mortar. Following this, 5 cm$^3$ of the sample was placed in a 50 ml beaker and 25 ml of 1M KCl solution was added. The suspension was mixed thoroughly and left until the next day, while protected from evaporation. The next day, the suspension was mixed again and the pH measured using a pH meter.

### Heavy metal determination

Heavy metal determination was performed in mineralized plant and soil samples using voltamperometry methods. Voltammetric measurements were performed using a MVA-22-894 Professional VA Automated Analyzer (Metrohm AG Ltd, Switzerland) with a three-electrode system consisting of a hanging mercury drop electrode (HMDE) as a working electrode, a

platinum (Pt) auxiliary electrode and an Ag/AgCl/ KCl (3 mol/l) reference electrode. Only ultrapure and Suprapur chemicals were used.

**Plant and soil sample preparation.** Accurately-weighed (circa 250 mg) air-dried *T. officinale* samples (flowers, leaves, roots) were shredded in a mortar, put into ceramic crucibles and mineralized in a muffle furnace according to the following time-temperature program: 60 min initial charring at 150˚C, 120 min burning at 300˚C, and 240 min combustion at 450˚C. After cooling the oven, the crucibles were transferred to an exsiccator, 2 ml $HNO_3$ (65%) was added to each sample and allowed to evaporate slowly at 120˚C. The samples were then subjected to 60 min calcination on a hot plate at 450˚C. Following this, 1 ml $HNO_3$ (65%) was added to the residue, and the mixture was subjected to slow evaporation at 120˚C and calcining (450˚C, 60 min). These operations were repeated until a light gray, easily digestible ash was obtained. The solid was then dissolved in 2 ml HCl (10%), transferred to 100 ml flasks and made up to the mark with ultra-pure water.

Accurately-weighed (circa 1 g) air-dried soil samples were shredded in a mortar and put into ceramic crucibles for mineralization. Following this, 7.5 ml HCl (30%) was added, followed by 2.5 ml $HNO_3$ (65%). In parallel, two blank tests were prepared with only acids in the crucibles. The next day, the samples were mineralized for three hours, initially at 100˚C then gradually increasing to 200˚C. When 30% of the acid mixture evaporated, the crucibles were allowed to cool and then supplemented with a 3:1 mixture of HCl (30%) and $HNO_3$ (65%). The extracts were transferred to 50 ml volumetric flasks, made up to the mark with ultra-pure water, then filtered through PTFE syringe filters (0.45 μm, FilterBio) into 100 ml flasks and made up to the mark with ultra-pure water. Suprapur acids were from Merck. Ultra-pure water was obtained from a Milli-Q Pure Water System.

**Determination of zinc, cadmium, lead, copper and thallium content by anodic stripping voltammetry (ASV) at hanging mercury dropping electrode (HMDE).** ASV is a two-step measurement. In the first step, the metal ions present in the test solution are deposited on the surface of a mercury electrode (amalgamation) at a deposition potential range of -1.15 V to -0.8 V. In the second step, all the deposited ions are anodically stripped by scanning the potential range from -1.15 to +0.1 V. All the measurements were performed by standard addition, in which the sample was first taken into the polarographic vessel and then the current was measured. After the addition of 0.1 ml of standard, the procedure was repeated two times and the current was measured. After all the measurements, an extrapolation curve was plotted between current versus concentration, and this was used to indicate the amount of metals present in the sample solution.

The Zn, Cd, Pb and Cu levels were determined using KCl-acetate buffer (pH 4.6) with a scan rate of 59.5 mV/s and pulse amplitude 50mV by HMDE standard addition. The solution was stirred during pre-electrolysis at -1.15 V (vs. Ag/AgCl) for 90 seconds and the potential was scanned from -1.15 V to +0.1 V (vs. Ag/AgCl). The KCl-acetate buffer was prepared from 5.59 g of Suprapur KCl, 2.5 ml of Suprapur acetic acid (100%), 10 ml of ultrapure water to which 2.5 ml of Suprapur ammonia (25%) was added slowly (to prevent heat generation) and the 50 ml volumetric flask was made up to the mark with ultra-pure water. Standard solution consisted of zinc (4 mg/l), cadmium (0.5 mg/l), lead (0.5 mg/l), and copper (2.5 mg/l) standards (all from Chempur) and was prepared from stock solutions of 1 g/l. All other reagents were from Merck.

Measurements: 2 ml of sample, 8 ml ultra-pure water and 1 ml of KCl-acetate buffer (pH 4.6) were placed in a polarographic vessel and then the measurement was started under the given parameters. After the sample voltammogram was recorded, 0.1 ml of mixed standard (Zn, Cu, Cd and Pb) was added twice and then the voltammogram of the standard was recorded. Operating parameters: working electrode HMDE, stirrer speed 2000 cm$^{-1}$,

deposition potential -1.15 V, deposition time 90 s, pulse amplitude 0.05 V, start potential -1.15 V, end potential 0.05 V, voltage step 0.006 V, voltage step time 0.1 s, sweep rate 0.06 V/s, Zn peak potential—0.98 V, Cd peak potential—0.56 V, Pb peak potential—0.38 V, Cu peak potential—0.10 V.

Thallium content was determined using ethylenediaminetetraacetic acid (EDTA) solution. To a 25 ml volumetric flask, 0.93g of EDTA (Merck) was added and dissolved in ultra-pure water. A standard solution of thallium (0.5 mg/l) was prepared from a 1g/l stock solution (Inorganic Ventures).

Measurements: 5 ml of sample, 5 ml ultra-pure water, 0.2 ml EDTA solution and 1 ml of KCl-acetate buffer were placed in a polarographic vessel. After the sample voltammogram was recorded, 0.1 ml of thallium standard was added twice, and the voltammogram of the standard was recorded. Operating parameters: working electrode HMDE, stirrer speed 2000 cm$^{-1}$, deposition potential -0.8 V, deposition time 180 s, pulse amplitude 0.05 V, start potential -0.8 V, end potential -0.2 V, voltage step 0.006 V, voltage step time 0.1 s, sweep rate 0.06 V/s, Tl peak potential -0.45 V.

**Determination of nickel and cobalt content by adsorptive cathodic stripping voltammetry (AdCSV) at hanging mercury dropping electrode (HMDE).** The supporting electrolyte (ammonia buffer) was prepared by slowly adding 11.25 ml of Suprapur ammonia (25%) to 5.3 ml of Suprapur hydrochloric acid (30%) and 10 ml of ultra-pure water into a 50 ml volumetric flask. The level was made up to the 50ml mark with ultra-pure water. Dimethylglyoxime (DMG) solution was prepared by dissolving 0.76 g of DMG in 25 ml of ultra-pure water. The standard solution consisted of nickel (1 mg/l) and cobalt (0.5 mg/l), both from Chempur, prepared from stock solutions of 1g/l. All other reagents were from Merck from Merck.

Measurements: 0.5 ml of sample, 0.1 ml DMG solution, 9.5 ml ultra-pure water and 1 ml of ammonia buffer were added to a polarographic vessel and the measurement was started under the given parameters. After the sample voltamogramme was recorded, 0.1 ml of mixed standard (Co and Ni) was added twice and then voltamogramme of the standard was recorded.

The following operating parameters were used: working electrode HMDE, stirrer speed 2000 cm$^{-1}$, deposition potential -700 mV, deposition time 90 s, pulse amplitude 0.05 V, start potential -0.6 V, end potential -1.25 V, voltage step 0.004 V, voltage step time 0.3 s, sweep rate 0.013 V/s, Co peak potential—1.13 V, Ni peak potential—0.97 V.

## Detection of PAHs, PBCs, dioxins and pesticides

Extraction of comminuted, air-dried soil and plant samples was performed three times (each 15 minutes) in an ultrasonic bath with a mixture of dichloromethane and acetone (9:1, v/v). The obtained extracts were pooled, dried over anhydrous sodium sulfate and then concentrated at 30°C in a rotary evaporator. About 1 mg of each extract was dissolved in 1 ml petroleum ether and uploaded to silica gel column (MN-Kieselgel 60, 1 cm x 6 cm); the PAHs and PCBs were separated into two fractions by elution with 8 and 4 ml of petroleum ether, respectively. The eluted PAH and PCB fractions were evaporated; about 1mg of each extract was dissolved in 100 μl of dichloromethane and analyzed by GC-MS.

Three calibration curves were made for each of 18 PAH, 12 PCB, 17 dioxin and 18 pesticide standards (S1 Table) to determine their retention times (RT), detection limits (LOD), precision and accuracy of the analyses by calculating the standard deviation, absolute and relative errors. Identification of individual compounds was made on the basis of characteristic mass spectra and NIST 11 library.

The analyses were carried out on a GCMS-QP2010 with a mass detector (Shimadzu). As a carrier gas, helium was used at a column head pressure of 65.2 kPa, with a DB-5 MS column

(Zebron, Phenomenex): thickness 0.25 μm, length 30 m, diameter 0.25 μm. The column oven temperature cycle started from 150°C; this temperature was held for three minutes then ramped to 310°C at 4°C/min; the final temperature was then held for 10 minutes. The ion source temperature was 200°C and the interface temperature was 310°C. Ionization 70 eV. The injection mode was split, and the split ratio was 10. Injection volume 1 μl. Quantitative determinations of the PAHs, PCBs, dioxins and pesticides in the plant and soil samples were made using the internal standard method. 2-methylanthracene was used as an internal standard for the determination of PAHs, and cholestane for the determination of other compounds. All standards, solvents and reagents were from Merck.

## Detection of mycotoxins

Accurately weighed samples (c.a. 0.5 g each, S2 Table) were mixed with 5 ml acetonitrile/water mixture (3:1 volume ratio), shaken for 40 minutes on a shaker at 250 RPM and centrifuged for 30 minutes at 3000 x g. Supernatants were twice degreased with 4 ml of hexane (15 min shaking) and water was removed from the collected acetonitrile/water phases by the addition of 0.2 g NaCl. The collected solutions, evaporated at 70°C, were dissolved in 15 ml of chloroform/ methanol mixture (9:1 volume ratio), sonicated and centrifuged (30 min, 3000 x g). The collected chloroform phases were dried under a nitrogen stream and re-dissolved in 2 ml of methanol or acetonitrile. They were then subjected to LC-MS analysis using LC-MS grade solvents from Merck. Prior to injection on the LC-MS system, the samples were filtered through a 4 mm, 0.2 μm polytetrafluoroethylene (PTFE) syringe filter (FilterBio).

Standards of 65 mycotoxins were purchased via the Axxora platform (http://www.axxora. com). The mycotoxins were separated by reverse-phase chromatography using a Shimadzu LC-MS 2020 system. The separation was achieved with a Kinetex EVO C18 column (2.6 μm particle size) preceded by a SecurityGuard™ ULTRA Holder pre-column, both supplied by Phenomenex. The applied mobile phase, ionization mode, characteristic ions and retention times of the mycotoxins are presented in S3 Table, while selected-ion monitoring (SIM) chromatograms of mycotoxins' standards are shown in S1 Fig. The following analysis parameters were used: injection volume 5 μl, flow 0.2 ml/min, detector voltage 4.5 kV, nebulizing gas flow 1.5 L/min, drying gas flow 15 L/min, scan speed 1500 u/sec, DL temperature 250 °C, heat block 250 °C, oven temperature 40 °C.

## Microbiological analyses

Accurately-weighed samples (circa 1 g for bacteriological analysis and circa 3 g for mycological analysis, each in three independent replications) were flooded with 100 ml of sterile 0.85% NaCl, shaken for 20 minutes at 25°C, and then 250 μl portions of each extract were sterile plated onto Columbia Agar plates with 5% sheep blood (CASB) and Sabouraud Glucose Selective Agar plates (SGSA), both from Becton Dickinson. Half of the inoculated CASB plates were kept aerobically, while the second half was kept anaerobically using the anaerobic jar (Roth) and AnaeroGen™ sachets (Thermo Scientific). Each inoculation variant was performed four times. All CASB plates were incubated at 37°C for 24 hours. Following this, the number of colony forming units (CFU) was determined using a SCAN 500 automatic bacterial colony counter (Interscience). Individual bacterial colonies were picked separately from each CASB plate, sieved onto new sterile CASB plates and cultured in the same way as the plates from which they were taken. After 24 hours of incubation, the bacteria were Gram stained (Gram Stain Kit, Thermo Scientific) and examined under a microscope to distinguish gram-positive and gram-negative colonies and bacterial morphology; this was a prerequisite for strain identification using an automated Vitek-2 system (Biomerieux). The procedure was repeated until

complete homogeneity of the bacterial cultures was achieved. Homogeneous bacterial colonies were applied to correctly-selected Vitek-2 disposable cards (Biomerieux) according to the producer's instructions; this allowed the identification of over 350 clinically-relevant bacteria and yeast.

Inoculated SGSA plates were kept at 25˚C for one to four weeks; following this, the cfus were counted. In order to obtain homogeneous fungal cultures, passages of the originally isolated colonies were performed several times. Fungi were identified macroscopically, based on the morphology of the grown colonies, and microscopically, based on preparations from isolated fungal colonies.

The Gram-stained bacteria and unstained fungi were subjected to phase-contrast microscope examination using a Axio Vert. A1 fluorescence inverted microscope (Zeiss) equipped with Axio Cam ICc 5 camera (Zeiss) and Zen lite 2012 software (Zeiss).

## Parasitological analyses

The soil and dandelion samples were checked for the presence of parasite eggs by the flotation technique [124]. Due to the wide range of the parasites included in the study, i.e. comprising nematodes, tapeworms, flukes and protozoa, a multiplex PCR assay was used to detect them: the test was based on the amplification of a short (250 bp) 18S rDNA fragment shared by flukes, nematodes, tapeworms and protozoa (S4 Table).

**DNA extraction and amplification.** DNA was extracted individually from air dried dandelion and soil samples using a Stool DNA Purification Kit (EURx, Poland) according to the manufacturer's protocol. The dried samples were rehydrated with MiliQ water and then the DNA was isolated according to the manufacturer's protocol.

Helminth DNA (positive control samples) was isolated from adult worms (*Toxocara cati*, *Toxocara canis*, *Echinococcus* spp., *Dipylidium caninum*, *Taenia* spp., *Fasciola hepatica*) using a Blood and Tissue DNA isolation kit (Qiagen) according to the manufacturer's protocol. DNA from the eggs of *Trichuris vulpis* (fox), *Toxocara cati* (cat), *Toxocara canis* (dog) and Oxyuridae sp. (turtle) was isolated using a Stool DNA Purification Kit (EURx, Poland). All DNA amplifications were performed using the DNA Engine T100 Thermal Cycler (BioRad). In some cases, nuclease-free water was added to the PCR mix instead of the tested DNA as a negative control. The PCR products were then visualized on a 1.2% agarose gel (Promega) stained with SimplySafe (EURx). Visualization was performed using ChemiDoc, MP Lab software (Imagine, BioRad).

**Helminth test.** 18S rDNA sequences of *Dipylidium caninum*, *Hymenolepis nana*, *Taenia solium*, *Taenia saginata*, *Taenia taeniaeformis*, *Echinococcus granulosus*, *Echinococcus multilocularis*, *Fasciola hepatica*, *Fascioloides magna*, *Ascaris lumbricoides*, *Ascaris suum*, *Enterobius vermicularis*, *Toxocara canis*, *Toxocara cati*, *Trichuris trichiura*, *Trichuris vulpis* were used to design a six-primer set. Four 18S rDNA sequences of *Giardia intestinalis* were used to design PCR primers. Detailed data on the parasites and 18S rDNA sequences are presented in S4 Table. Partial regions (250bp) of 18S rDNA of all tested helminths were amplified using the following sets of primers: CF1 (5′ `GCGGGGRCGTTTGTATGGCTGC` 3′), NF1 (5′ `ACGGGG RCATTCGTATyGCTGC` 3′), TtF 1 (5′ `GCGGGGACGTTTGTATGGTTGCG` 3′), FhF 1 (5′ `ACGGGGGCATTTGTATGGCGGT` 3′), CNR1 (5′ `CAACCATACTTCCCCCGGAACCSAAA` 3′) and TR1 (5′ `CCATACTTCCCCCGGAGCCCAAA` 3′). PCR was performed in a TM 100 (BioRad) thermal cycler. The reactions were conducted in a 50 μl reaction mixture containing 5.0 μl of DNA template, 1 μl (1U) of Color Taq DNA Polymerase (EURx), 1 μl of dNTPs mix (10 mM), 0.5 μl of TR1, TtF1 and FhF1, 1 μl of CF1 and CNR1, 1.5 μl of NF1 primer (20 mM), 5 μl of 10 × Polymerase buffer (pH 8.6, 25 mM $MgCl_2$) and 33.0 μl of MiliQ water. A negative

control—nuclease-free water was added to the PCR mix instead of the tested DNA. DNA amplification was performed according to the following program: denaturation at 95°C for 1 min, followed by 34 cycles of denaturation at 95°C for 10 s, annealing at 58°C for 10 s and extension at 72°C for 20 s, with a final extension performed at 72°C for 3 min.

**Giardia test.** Partial regions of 18S rDNA were amplified using the following sets of primers: GF1 (5′ TCCGGTCGATCCTGCCGGA 3′) and G452R (5′ GCTGCTGGCACCAGACC TTG 3′). PCR was performed in a TM 100 (BioRad) thermal cycler. The reactions were conducted in a 50 μl reaction mixture containing 5.0 μl template DNA, 1.0 μl (1U/ μl) of Color Taq DNA Polymerase (EURx), 1 μl of dNTPs mix (10 mM), 0.5 μl of GF1 and G452R primer (20 mM), 5 μl of 10 × Polymerase buffer (pH 8.6, 25 mM MgCl$_2$) and 37.0 μl of MiliQ water. A negative control—nuclease free water was added to the PCR mix instead of the tested DNA. DNA amplification was performed using the DNA Engine T100 Thermal Cycler (BioRad) according to the following program: denaturation at 95°C for 1 min, followed by 34 cycles of denaturation at 95°C for 15 s, annealing at 55°C for 15 s and extension at 72°C for 30 s, with a final extension performed at 72°C for 3 min.

**DNA sequencing.** Giardia positive dandelion samples were used to amplify the triosephosphate isomerase (TPI) gene fragment using the tpi nested PCR protocol [125]. The selected PCR products were purified with QIAquick Gel Extraction Kit (Qiagen, Germany) and then were sequenced directly using ABI BigDye™ chemistry (Applied Biosystems, USA) on an ABI Prism 373xl or an ABI Prism 3100™ automated sequencer (Genomed, Warsaw, Poland). The results were compared with relevant sequences from the GenBank database.

## Statistical analyses

The results of the voltamperometric measurements and the cfu determinations were subjected to statistical analysis. The normality of their distribution was checked using the Kolmogorov-Smirnov (K-S) test. As the distributions were normal, the Student's t-test was used to compare them. A significance level of 95% was assumed (p<0.05). STATISTICA 6.1 software (StatSoft Polska) was used for all statistical testing.

## Supporting information

**S1 Fig. Selected-ion monitoring chromatograms of mycotoxins' standards.**
(TIF)

**S1 Table. List of standards used, their retention times (RT) and limits of detection (LOD).**
(DOCX)

**S2 Table. Extraction of mycotoxins from *Taraxacum officinale* and soil samples.**
(DOCX)

**S3 Table. Summary of LC-MS detection conditions of 65 mycotoxins.**
(DOCX)

**S4 Table. List of 18S rDNA sequences used to design PCR reaction primers (1–21) and parasites used as control samples (22–35).**
(DOCX)

**S5 Table. Correlations between the concentrations of metals measured in individual parts of plants and soil samples.**
(XLSX)

**S6 Table. Raw data: (A) concentration of heavy metals, (B) soil pH.**
(XLSX)

**S7 Table. Raw data: Microorganisms isolated from *Taraxacum officinale* and soil.**
(XLSX)

## Acknowledgments

The authors would like to thank Dr. Justyna Sobich for reading the work and valuable comments.

## Author Contributions

**Conceptualization:** Mieczysława Irena Boguś.

**Formal analysis:** Mieczysława Irena Boguś.

**Funding acquisition:** Mieczysława Irena Boguś.

**Investigation:** Anna Katarzyna Wrońska, Agata Kaczmarek, Mikołaj Drozdowski, Zdzisław Laskowski, Anna Myczka, Aleksandra Cybulska, Marek Gołębiowski, Adrianna Chwir-Gołębiowska, Lena Siecińska, Ewelina Mokijewska.

**Methodology:** Mieczysława Irena Boguś, Anna Katarzyna Wrońska, Agata Kaczmarek, Zdzisław Laskowski, Marek Gołębiowski.

**Supervision:** Mieczysława Irena Boguś.

**Validation:** Marek Gołębiowski.

**Visualization:** Mieczysława Irena Boguś.

**Writing – original draft:** Mieczysława Irena Boguś.

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
