## [Decision Letter · Decision Letter 0]

31 Oct 2022

PONE-D-22-12598A comprehensive analysis of chemical and biological pollutants (natural and anthropogenic origin)  of soil and dandelion (Taraxacum officinale) samples -  pathogenic microorganisms and parasites pose a greater threatPLOS ONE

Dear Dr. Boguś,

Thank you for submitting your manuscript to PLOS ONE. After careful consideration, we feel that it has merit but does not fully meet PLOS ONE’s publication criteria as it currently stands. Therefore, we invite you to submit a revised version of the manuscript that addresses the points raised during the review process.

We look forward to receiving your revised manuscript.

Kind regards,

Estibaliz Sansinenea

Academic Editor

PLOS ONE

Journal Requirements:

2. In your Methods section, please provide additional information regarding the permits you obtained to collect samples for the present study. Please ensure you have included the full name of the authority that approved the field site access and, if no permits were required, a brief statement explaining why.

“This work was supported by the Marshal’s Office of the Mazowieckie Voivodeship grant RPMA.01.02.00-14-5626/16 to the Biomibo company. There was no additional external funding received for this study. Biomibo covered the cost of the salaries of its employees (MIB, AC-G, LS, EM), provided support in the purchase of chemicals, and made laboratory equipment available for all authors. The specific roles of the authors are articulated in the ‘author contributions’ section. The funder did not has any additional role in the study design, data collection, analysis and interpretation, decision to publish, or preparation of the manuscript.”

Additional Editor Comments (if provided):

The reviewers have commented about several issues that need to be attended. I recommend a major revision of this MS.

Reviewers' comments:

Reviewer's Responses to Questions

**Comments to the Author**

1. Is the manuscript technically sound, and do the data support the conclusions?

Reviewer #1: No

Reviewer #2: Yes

2. Has the statistical analysis been performed appropriately and rigorously? 

Reviewer #1: Yes

Reviewer #2: Yes

3. Have the authors made all data underlying the findings in their manuscript fully available?

Reviewer #1: No

Reviewer #2: Yes

4. Is the manuscript presented in an intelligible fashion and written in standard English?

Reviewer #1: Yes

Reviewer #2: Yes

5. Review Comments to the Author

Reviewer #1: A comprehensive analysis of chemical and biological pollutants (natural and anthropogenic origin) of soil and dandelion (Taraxacum officinale) samples - pathogenic microorganisms and parasites pose a greater threat

Although the title and purpose of this manuscript is very interesting, I think that in some cases the correct scientific method was not used.

Laboratory methods used for microorganisms are not very clear. Meanwhile, organisms like Francisella grow in specific medium and microaerophilic conditions.

Reviewer #2: The present work is a good effort to detect chemical and biological contaminations in soil and dandelion samples using a wide range of analytical methods (GC-MS, LC-MS, voltammetry, 505 microbiological and microscopic techniques, PCR). I agree with its publication. However, herewith I have few suggestions for its improvement

• Revise the title for clarity

• Make abstract and introduction short

• Discussion section needs improvement with more relevant latest articles

• Avoid use of pronouns (e.g. we, our etc.,) in the text

• Most of the headings are too lengthy. It would be better to shorten headings

• Increase visibility of the figures for clarity

• Supplementary data has captions while captions of main figures are missing

6. PLOS authors have the option to publish the peer review history of their article (what does this mean?). If published, this will include your full peer review and any attached files.

Reviewer #1: No

Reviewer #2: **Yes: **Muhammad Imran

---

## [Author Response · Author response to Decision Letter 0]

8 Jan 2023

Response to Reviewers

Comments to the Author

1. Is the manuscript technically sound, and do the data support the conclusions?

Reviewer #1: No

Reviewer #2: Yes

2. Has the statistical analysis been performed appropriately and rigorously?

Reviewer #1: Yes

Reviewer #2: Yes

3. Have the authors made all data underlying the findings in their manuscript fully available?

Reviewer #1: No 

Answer: After addition two supplementary tables (Supplementary Table 6 and 7) containing raw measurements all data underlying the findings presented in the manuscript are fully available. 

Reviewer #2: Yes

4. Is the manuscript presented in an intelligible fashion and written in standard English?

Reviewer #1: Yes

Reviewer #2: Yes

5. Review Comments to the Author

Reviewer #1: A comprehensive analysis of chemical and biological pollutants (natural and anthropogenic origin) of soil and dandelion (Taraxacum officinale) samples - pathogenic microorganisms and parasites pose a greater threat

Although the title and purpose of this manuscript is very interesting, I think that in some cases the correct scientific method was not used.

Laboratory methods used for microorganisms are not very clear. Meanwhile, organisms like Francisella grow in specific medium and microaerophilic conditions. 

Answer: It is difficult to respond to such a general comment because Reviewer #1 did not mention any specific methodological errors that he believed we had made. In this work, very diverse methods were used, from voltammetric determinations of heavy metals through gas and liquid chromatography coupled with mass spectroscopy (GC-MS and LC-MS), microbiological methods and PCR. All these methods are described in detail in the Materials and Methods section. We expect Reviewer #1 to present specific objections to the methods used by us.

In microbiological studies, we used classical inoculation methods on commercially available media: Columbia Agar plates with 5% sheep blood (CASB) and Sabouraud Glucose Selective Agar plates (SGSA), both from Becton Dickinson. Half of the inoculated CASB plates were kept aerobically, while the second half was kept anaerobically using the anaerobic jar (Roth) and AnaeroGen ™ sachets (Thermo Scientific). All CASB plates were incubated at 37oC for 24 hours. Following this, the number of colony forming units (CFU) was determined using a SCAN 500 automatic bacterial colony counter (Interscience). The bacteria were Gram stained (Gram Stain Kit, Thermo Scientific) and examined under a microscope to distinguish gram-positive and gram-negative colonies and bacterial morphology as this was a prerequisite for strain identification using an automated Vitek-2 system (Biomerieux). Homogeneous bacterial colonies were applied to correctly-selected Vitek-2 disposable cards. The modern, Vitek-2 automated microbiology system is validated and widely used in clinical microbiology to identify various bacterial strains in diagnostic laboratories and hospitals around the world (https://www.biomerieux-usa.com/vitek-2;
https://www.ncbi.nlm.nih.gov/pmc/articles/PMC254354/ ). The composition of the growing media used in the Vitek-2 system cards dedicated various groups of microorganisms is a manufacturer's secret.

Francisella tularensis might be grown in defined media such as Chamberlains or in non-selective media such as Mueller-Hinton broth and a modified Brain Heart Infusion (BHI) broth enhances the growth of this microbe (Morris BJ, Buse HY, Adcock NJ, Rice EW. A novel broth medium for enhanced growth of Francisella tularensis. Lett Appl Microbiol. 2017 Jun;64(6):394-400. doi: 10.1111/lam.12725). We do not know which medium is used in the Biomerieux cards for Francisella tularensis identification (manufacturer secret).

Reviewer #2: The present work is a good effort to detect chemical and biological contaminations in soil and dandelion samples using a wide range of analytical methods (GC-MS, LC-MS, voltammetry, 505 microbiological and microscopic techniques, PCR). I agree with its publication. However, herewith I have few suggestions for its improvement

• Revise the title for clarity 

Answer: Revised title: “A comprehensive analysis of chemical and biological pollutants (natural and anthropogenic origin) of soil and dandelion (Taraxacum officinale) samples.”

• Make abstract and introduction short 

Answer: The abstract is only 268 words long and is written in a very concise manner. Attempts to shorten it result in a loss of clarity. The Introduction has been shortened in such a way as not to lose important information and clarity of the message, as well as the justification why the research described in this work was carried out.

• Discussion section needs improvement with more relevant latest articles 

Answer: Done

• Avoid use of pronouns (e.g. we, our etc.,) in the text 

Answer: Done

• Most of the headings are too lengthy. It would be better to shorten headings 

Answer: Where possible, subsection titles have been made shorten.

• Increase visibility of the figures for clarity 

Answer: Unfortunately, improving the quality of the Figure 2 and Supplementary Figure 1 is not possible for technical reasons. The quality of the chromatograms depends on the factory software of the GC-MS and LC-MS chromatographs (Shimadzu). The quality of Figures 1, 3, 4 and 5 seems technically satisfactory.

• Supplementary data has captions while captions of main figures are missing 

Answer: The titles of the main figures are placed in the main text according to the editorial requirements of PlosOne i.e. Figure 1 lines 130-134, Figure 2 lines 222-224, Figure 3 lines 249-250, Figure 4 lines 301-302, Figure 5 lines 319-329, respectively.

---

## [Editor Report · Decision Letter 1]

10 Jan 2023

A comprehensive analysis of chemical and biological pollutants (natural and anthropogenic origin)  of soil and dandelion (Taraxacum officinale) samples

PONE-D-22-12598R1

Dear Dr. Boguś,

We’re pleased to inform you that your manuscript has been judged scientifically suitable for publication and will be formally accepted for publication once it meets all outstanding technical requirements.

Kind regards,

Estibaliz Sansinenea

Academic Editor

PLOS ONE

Additional Editor Comments (optional):

The authors have followed all.comments improving MS therefore it can be accepted in the current form.
---

## [Editor Report · Acceptance letter]

11 Jan 2023

PONE-D-22-12598R1 

A comprehensive analysis of chemical and biological pollutants (natural and anthropogenic origin)  of soil and dandelion (*Taraxacum officinale*) samples. 

Dear Dr. Boguś:

I'm pleased to inform you that your manuscript has been deemed suitable for publication in PLOS ONE. Congratulations! Your manuscript is now with our production department. 

Kind regards, 

on behalf of

Dr. Estibaliz Sansinenea 

Academic Editor

PLOS ONE